# Mitochondrial Genome Fragmentation Occurred Multiple Times Independently in Bird Lice of the Families Menoponidae and Laemobothriidae

**DOI:** 10.3390/ani13122046

**Published:** 2023-06-20

**Authors:** Yalun Dong, Martina Jelocnik, Amber Gillett, Ludovica Valenza, Gabriel Conroy, Dominique Potvin, Renfu Shao

**Affiliations:** 1Centre for Bioinnovation, University of the Sunshine Coast, 90 Sippy Downs Drive, Sippy Downs, QLD 4556, Australia; yad002@student.usc.edu.au (Y.D.); mjelocni@usc.edu.au (M.J.); gconroy@usc.edu.au (G.C.); 2School of Science, Technology and Engineering, University of the Sunshine Coast, 90 Sippy Downs Drive, Sippy Downs, QLD 4556, Australia; 3Australia Zoo Wildlife Hospital, 1638 Steve Irwin Way, Beerwah, QLD 4519, Australia; amber@wildlifewarriors.org.au (A.G.); ludov@wildlifewarriors.org.au (L.V.); 4School of Science, Technology and Engineering, University of the Sunshine Coast, 1 Moreton Parade, Petrie, QLD 4502, Australia; dpotvin@usc.edu.au

**Keywords:** mitochondrial genome fragmentation, Menoponidae, Laemobothriidae, phylogeny

## Abstract

**Simple Summary:**

To understand mitochondrial genome fragmentation in bird lice, we sequenced the mitochondrial genomes of 17 species of bird lice in the families Menoponidae and Laemobothriidae. Of the 17 species, four species of Menoponidae have fragmented mitochondrial genomes, whereas the other 13 species of both Menoponidae and Laemobothriidae retain the typical single-chromosome mitochondrial genomes. Our phylogenetic analyses showed that mitochondrial genome fragmentation occurred multiple times independently in these two families. We also found derived mitochondrial minichromosomal characters shared between bird lice in the same genus and among different genera. We conclude that while mitochondrial genome fragmentation as a general feature does not unite all the parasitic lice that have this feature, each independent mitochondrial genome fragmentation event may produce shared derived mitochondrial minichromosomal characters that can be useful for resolving the phylogeny of parasitic lice at different taxonomic levels.

**Abstract:**

Mitochondrial (mt) genome fragmentation has been discovered in all five parvorders of parasitic lice (Phthiraptera). To explore whether minichromosomal characters derived from mt genome fragmentation are informative for phylogenetic studies, we sequenced the mt genomes of 17 species of bird lice in Menoponidae and Laemobothriidae (Amblycera). Four species of Menoponidae (*Actornithophilus* sp. 1 ex [pied oystercatcher], *Act.* sp. 2 ex [masked lapwing], *Austromenopon* sp. 2 ex [sooty tern and crested tern], *Myr.* sp. 1 ex [satin bowerbird]) have fragmented mt genomes, whereas the other 13 species retain the single-chromosome mt genomes. The two *Actornithophilus* species have five and six mt minichromosomes, respectively. *Aus.* sp. 2 ex [sooty tern and crested tern] has two mt minichromosomes, in contrast to *Aus.* sp. 1 ex [sooty shearwater], which has a single mt chromosome. *Myr.* sp. 1 ex [satin bowerbird] has four mt minichromosomes. When mapped on the phylogeny of Menoponidae and Laemobothriidae, it is evident that mt genome fragmentation has occurred multiple times independently among Menoponidae and Laemobothriidae species. We found derived mt minichromosomal characters shared between *Myrsidea* species, between *Actornithophilus* species, and between and among different ischnoceran genera, respectively. We conclude that while mt genome fragmentation as a general feature does not unite all the parasitic lice that have this feature, each independent mt genome fragmentation event does produce minichromosomal characters that can be informative for phylogenetic studies of parasitic lice at different taxonomic levels.

## 1. Introduction

Parasitic lice (infraorder Phthiraptera) are obligate parasites living on birds and mammals [1,2,3]. Among parasitic lice, chewing lice are in four parvorders: Amblycera, Ischnocera, Rhynchophthirina, and Trichodectera, while sucking lice are in the parvorder Anoplura [3]. Fragmented mitochondrial (mt) genomes have been found in 21 species of Anoplura [4,5,6,7,8,9,10,11,12,13,14], one species of Rhynchophthirina [15], and five species of Trichodectera [16,17]. Each of these species has 9 to 20 minichromosomes, which are usually less than 4 kb in size. Cameron et al. [18] and three recent studies showed that mt genome fragmentation also occurred in species of Ischnocera [19,20] and Amblycera [21]. Cameron et al. [18] reported five species of Ischnocera with fragmented mt genomes based on PCR tests and partial mt genome sequences. Sweet et al. [19] reported that the mt genomes of three *Columbicola* species of dove lice (Ischnocera) each had 15–17 minichromosomes that were less than 4 kb in size. Sweet et al. [20] reported that another 14 species of Ischnocera in 14 different genera had fragmented mt genomes; each of these species had 3 to 10 minichromosomes ranging from around 1 kb to 11 kb in size. Sweet et al. [21] reported that the mt genomes of four species of Amblycera from different genera (*Cummingsia*, *Laemobothrion, Macrogyropus*, and *Myrsidea*) each had three or seven minichromosomes.

Mitochondrial genome fragmentation occurred at least nine times independently in the parvorder Ischnocera and at least three times independently in the parvorder Amblycera according to Sweet et al. [20]. These fragmentation events are independent of the fragmentation observed in eutherian mammal lice of the parvorders Anoplura, Rhynchophthirina, and Trichodectera, which apparently occurred only once in the most recent common ancestor of these three parvorders [3,16,19,20]. The typical single-chromosome mt genome has not been seen in any species from Anoplura, Rhynchophthirina, or Trichodectera, but has been seen in 10 species from 10 different genera of Ischnocera and seven species of Amblycera from three families (Boopiidae; Menoponidae, and Ricinidae) [16,21,22,23].

In the current study, we sequenced the mt genomes of 17 species of amblyceran lice: 14 from the family Menoponidae and three from the family Laemobothriidae (Table 1). Our aims are to investigate: (1) how widespread mt genome fragmentation occurred in the Amblycera, using Menoponidae as an example; (2) whether other species in Laemobothriidae have fragmented mt genomes as reported for *Laemobothrion (Laemobothrion) tinnunculi* (Linnaeus, 1758) [21]; and (3) whether minichromosomal characters derived from mt genome fragmentation are informative for resolving the phylogeny of parasitic lice at low taxonomic levels such as genus and species.

## 2. Materials and Methods

### 2.1. Collection of Bird Lice

We collected 17 species of bird lice; 16 of these species were collected in the Australian Zoo Wildlife Hospital (AZWH) from euthanized birds and one species was from live birds (Australian white ibis, identified by D.P.) captured at the UniSC Fraser Coast campus (Table 1). Euthanized birds were initially identified by AZWH veterinary staff and verified by two of the authors (Y.D. and R.S.) based on morphology according to Pizzey et al. [24] and Menkhorst et al. [25]. We used the post-mortem-ruffling method to collect bird lice [26]. Ethyl acetate was used to treat euthanized birds if lice were still alive. Ethanol-acetate-soaked cotton balls were put in sealed bags containing euthanized birds, one bird per bag to avoid contamination. After 5 to 10 min, treated birds were placed on a clean A1-size white paper, with their feathers ruffled to get lice off bird feathers onto the white paper. Lice were kept in 2-mL collection tubes filled with 80% ethanol at −80 °C in a freezer until DNA extraction. Lice were checked under a microscope (Nikon SMZ 800N), imaged, and identified by morphological features and host records according to the world checklist of chewing lice [1].

### 2.2. DNA Extraction, Mitochondrial Genome Sequencing, and Assembly

The genomic DNA of all of the 17 bird louse species (20 samples in total; Table 1) were extracted with the DNeasy Blood and Tissue kit (QIAGEN); 1 to 50 individual lice of each sample (depending on the body size of the louse) were used in DNA extraction. For the three species of *Laemobothrion*, we used single individual lice for extraction; for the species of Menoponidae, we used 14 to 50 individual lice for DNA extraction. Genomic DNA was checked by Nanodrop and Qubit to meet quality and quantity requirements for sequencing by Novogene (HK) with Illumina Novaseq 6000 platform (PE 150). According to Novogene (HK), the genomic DNA was randomly sheared into short fragments (350 bp); the obtained fragments were end-repaired, A-tailed, and ligated with Illumina adapters. The fragments with adapters were PCR amplified, size selected, and purified; the fragments from each genomic DNA sample formed a sequencing library. The library was checked with Qubit and real-time PCR for quantification and with Agilent Bioanalyzer for size distribution. Quantified libraries were sequenced on Illumina platforms. Then, 9.68 to 25.68 million cleaned Illumina sequence reads were obtained from each genomic DNA sample; these reads were deposited in the NCBI SRA database (Table 1). Illumina sequence reads were assembled using Geneious [27]. The Geneious assembler and medium sensitivity option were chosen in de novo assembly. The Geneious mapper and custom sensitivity option were chosen in reference mapping with the following parameters: (1) 80 bp minimum overlap and 95% minimum identity; (2) maximum 5% gaps per read with 2 bp maximum gap size; and (3) maximum mismatches per read 5% and maximum ambiguity 2. Crampton-Platt et al. [28] reported that the proportion of mt genome reads ranged from 0.5% to 1.4% in whole genome shotgun sequence reads; we thus used 1% of total sequence reads (around 100,000 to 200,000 reads) obtained from each genomic DNA sample as the initial seed dataset for de novo assembly. On average, each initial seed dataset had around 1000 to 2000 mt genome reads (150 bp each read), which were sufficient for us to obtain contigs covering multiple mt genes. The consensus sequences of the contigs generated by de novo assembly were searched with BLAST tools (blastn and blastx) in NCBI databases using the default parameters to identify mt gene sequence matches [29]. The consensus sequences that significantly matched (E-value < 10^−18^) to the mt gene sequences of amblyceran chewing lice were used as initial reference sequences for mapping with the total amount of Illumina sequence reads of each louse sample. For species with the typical single-chromosome mt genomes, repeatedly extending the contigs obtained from the initial references led to the assembly of the full-length mt genomes. For species with fragmented mt genomes, this same approach led to the assembly of multiple mt minichromosomes. We annotated these minichromosomes to identify the mt genes in each minichromosome. We then reiterated the process described above from de novo assembly to minichromosome annotation to target unidentified mt genes until no additional mt minichromosomes or genes could be identified. Protein-coding genes and rRNA genes were identified and annotated based on BLAST search results of GenBank [29]; tRNA genes were identified and annotated based on secondary structures produced by ARWEN [30] and tRNAscan-SE [31].

### 2.3. PCR Verification of Mitochondrial Minichromosomes

For the four species of Menoponidae with fragmented mt genomes (*Act.* sp. 1 ex [pied oystercatcher], *Act.* sp. 2 ex [masked lapwing], *Aus.* sp. 2 ex [sooty tern and crested tern], and *Myr.* sp. 1 ex [satin bowerbird]), we designed 17 pairs of specific outward primers to verify the circular organization and the size of the 17 minichromosomes obtained from Illumina sequence-read assembly. These primers were designed from the protein-coding or rRNA gene sequences of each minichromosome (Appendix A). The forward and reverse primers in each pair were next to each other with a small gap (200–400 bp) in between; PCRs with these primers amplified each circular minichromosome at full length except for the small gap between the two primers. The amplicons from each minichromosome were individually sequenced using the Novaseq 6000 platform as described above. The PCR setting was: 12.5 μL PrimeSTAR Max Premix (2X), 1 μL forward primer, 1 μL reverse primer, 1 μL DNA template, 9.5 μL ultrapure water (PCR grade). The PCR cycling condition was: 94 °C for 1 m; 35 cycles of 98 °C for 10 s, 60 °C for 15 s, 72 °C for 1 m; 72 °C for 10 m. Three negative controls were set up for each pair of PCR primers: (1) 1 μL ultrapure water (PCR grade) replacing the DNA template; (2) 1 μL ultrapure water replacing the forward primer; and (3) 1 μL ultrapure water replacing the reverse primer. PCR amplicons were checked by agarose gel (1%) electrophoresis and were sequenced using the Novaseq 6000 platform (PE 150) at Novogene (HK) with the same library construction methods as described above; the Illumina sequence reads were assembled with the same approach described above for genomic DNA sequence assembly.

### 2.4. Phylogenetic Analysis

We constructed phylogenetic trees of 23 species of bird lice: 19 of these lice in the family Menoponidae and the other 4 species in the family Laemobothriidae (Table 1). Sequences of 13 mt protein-coding genes (*atp6*, *atp8*, *cob*, *cox1*, *cox2*, *cox3*, *nad1*, *nad2*, *nad3*, *nad4*, *nad4L*, *nad5*, and *nad6*) were used in our phylogenetic analysis. The deduced amino acid sequences of these genes were individually aligned using the MAFFT algorithm implemented in the TranslatorX online platform [32]. Poorly aligned sites were removed by Gblocks vo.91b [33] with default parameters and gaps not allowed. For maximum likelihood (ML) and Bayesian inference (BI) analyses, we used partitions generated with PartitionFinder2 [34]. The ML tree was constructed with IQ-TREE [35] with 1000 ultrafast bootstrap replicates. The BI tree was constructed with MrBayes 3.2 6 [36]. MCMC generations were set as 5,000,000 and sample-frequency was set as every 1000 generations. We set a burn-in of 500,000 generations and the MCMC convergence was checked with Tracer v1.7.2 [37]. The ML tree and the BI tree were edited in Figtree v1.4.3 (http://tree.bio.ed.ac.uk/software/figtree) (accessed on 9 June 2023). We inferred the ancestral states of mt genome organization on the ML tree using Mesquite (Version 3.81) [38] using the likelihood ancestral state reconstruction method. We used both the Mk1 model (Markov 1 parameter) and the AsymmMk model (2 parameters) supported by Mesquite.

## 3. Results

### 3.1. Bird Lice Have Both Fragmented and Typical Single-Chromosome Mitochondrial Genome Organization

Among the 17 species of bird lice we sequenced for the first time in this study, 13 of them (10 species of Menoponidae and three species of Laemobothriidae) have the typical single-chromosome mt genomes (Figure 1A, Appendix A). The other four species of Menoponidae (*Actornithophilus* sp. 1 ex [pied oystercatcher], *Act.* sp. 2 ex [masked lapwing], *Austromenopon* sp. 2 ex [sooty tern and crested tern], and *Myrsidea* sp. 1 ex [satin bowerbird]) have fragmented mt genomes: (1) *Act.* sp. 1 ex [pied oystercatcher] have five minichromosomes and *Act.* sp. 2 ex [masked lapwing] have six minichromosomes (Figure 2); (2) *Aus.* sp. 2 ex [sooty tern and crested tern] has two minichromosomes (Figure 1B); and (3) *Myr.* sp. 1 ex [satin bowerbird] has four minichromosomes (Figure 3). All of the minichromosomes of these four species, except one minichromosome of *Aus.* sp. 2 ex [sooty tern and crested tern] (M1 minichromosome, 9499 bp in size, more details below), were verified by PCR (Appendix A). The failure to amplify the M1 minichromosome of *Aus.* sp. 2 ex [sooty tern and crested tern] by PCR was most likely due to its large size. The PCR amplicons from each minichromosome were also individually sequenced and assembled to verify the minichromosomes obtained from genomic DNA sequence assembly. The mean coverage of the mt minichromosomes identified ranges from 422 to 4396 (Appendix A). Our sequence comparison showed that *Franciscoloa* sp. 3 ex [pheasant coucal] is the same species as *F.* sp. 3 ex [little corella and galah]. All of the mt protein-coding genes and rRNA genes of *F.* sp. 3 ex [pheasant coucal] (GenBank accession numbers: ON303709) have 99.0–99.9%% sequence identity with their homologous genes of *F.* sp. 3 ex [little corella and galah] (*cox1* gene has 99.6% and 99.7% identity) (GenBank accession numbers: ON303707 and ON303708). The mt genomes of the 17 species of bird lice sequenced in the current study were annotated and deposited in GenBank (accession numbers listed in Table 1).

### 3.2. The Typical Single-Chromosome Mitochondrial Genome Was Retained in *Austromenopon* sp. 1 ex [Sooty Shearwater] but Broke Up into Two Chromosomes in *Austromenopon* sp. 2 ex [Sooty Tern and Crested Tern]

We assembled the mt genomes of *Aus.* sp. 1 ex [sooty shearwater] and *Aus.* sp. 2 ex [sooty tern and crested tern]. All of the 37 typical mt genes were identified in both species of *Austromenopon* (Figure 1). The 37 mt genes of *Aus.* sp. 1 ex [sooty shearwater] are on a single chromosome, 14,992 bp in size, which is typical of bilateral animals [39,40]. There are two non-coding regions in the mt genome of *Aus.* sp. 1 ex [sooty shearwater]: one is 127 bp in size between *cox1* and *atp6*, and the other is 327 bp between *trnM* and *trnD* (Figure 1). *Aus.* sp. 2 ex [sooty tern and crested tern] share six derived gene clusters (20 genes in total) with *Aus.* sp. 1 ex [sooty shearwater]: (1) *cox2-cox1-atp6-atp8*, (2) *nad2-nad5-I*, (3) *nad3-Y*, (4) *rrnL-rrnS-L_1_*, (5) *nad4L-nad4-E*, and (6) *P*-*K*-*S_1_* (Figure 1, Appendix A). However, the 37 mt genes of *Aus.* sp. 1 ex [sooty shearwater] are on two minichromosomes: M1 is 9499 bp in size with 16 genes, and M2 is 4943 bp with 21 genes (Figure 1, Appendix A). There are two non-coding regions in each minichromosome: for M1, one is 50 bp in size between *cox1* and *atp6*, and the other is 49 bp between *nad2* and *atp8*; for M2, one is 141 bp in size between *trnS1* and *trnQ*, and the other is 80 bp between *trnQ* and *nad4L* (Figure 1).

### 3.3. Mitochondrial Karyotype Differs between Two *Actornithophilus* Species

We assembled the mt genomes of *Act.* sp. 1 ex [pied oystercatcher] and *Act.* sp. 2 ex [masked lapwing], and identified 35 of the 37 typical mt genes in both species (*trnP* and *trnR* not found). These 35 genes are on five minichromosomes in *Act.* sp. 1 ex [pied oystercatcher], but are on six minichromosomes in *Act.* sp. 2 ex [masked lapwing] (Figure 2). The five minichromosomes (M1-M5) of *Act.* sp. 1 ex [pied oystercatcher] range from 2005 bp to 6112 bp (Figure 2, Appendix A). The smallest minichromosome, M1, has three genes, while the largest minichromosome, M5, has fifteen genes. The six minichromosomes (M1–M6) of *Act.* sp. 2 ex [masked lapwing] range from 1698 bp to 4361 bp (Figure 2, Appendix A). The smallest minichromosome, M1, has three genes, while the largest minichromosome, M6, has 12 genes.

*Actornithophilus* sp. 1 ex [pied oystercatcher] and *Act.* sp. 2 ex [masked lapwing] share obvious similarity in mt karyotype between each other, but not with any other parasitic lice (Figure 2): (1) both species have an *E-nad4L-nad4* minichromosome (i.e., M1); and (2) both species have five derived gene clusters in different minichromosomes (*V-K-cob-nad1*, *nad6-H-S2*, *nad2-S1-F*, *Y-atp8-atp6-N* and *G-I-cox2-cox1-C-cox3*) (Appendix A). These two species of *Actornithophilus*, however, are distinct from each other in mt karyotype: (1) *Act.* sp. 1 ex [pied oystercatcher] has five minichromosomes, while *Act.* sp. 2 ex [masked lapwing] has six minichromosomes; and (2) all of their relatable minichromosomes differ in gene content, with the only exception being the M1 minichromosome (Figure 2). The difference between *Act.* sp. 1 ex [pied oystercatcher] and *Act.* sp. 2 ex [masked lapwing] in mt karyotype can be accounted for by multiple events that occurred after the divergence of these two species from their most recent common ancestor: (1) either the split of M2 minichromosome of *Act.* sp. 1 ex [pied oystercatcher] into two minichromosomes, or the merger of M2 and M3 minichromosomes of *Act.* sp. 2 ex [masked lapwing] as one; (2) translocation of six genes (*trnQ*, *trnW*, *trnL1*, *nad5*, *trnD*, and *trnL2*) between different minichromosomes in either of the two species; and 3) inversion (i.e., change in transcription orientation) of *trnA*, *trnQ*, and *trnW* in either of the two species (Figure 2).

### 3.4. The Mitochondrial Genome of *Myrsidea* sp. 1 ex [Satin Bowerbird] Comprises at Least Four Minichromosomes

We assembled the mt genome of *Myr.* sp. 1 ex [satin bowerbird] and identified 23 of the 37 typical mt genes (13 mt protein-coding genes, two rRNA genes, and eight tRNA genes); the other 14 tRNA genes were not identified (Figure 3). These 23 mt genes are on four minichromosomes (M1–M4) ranging from 1969 bp to 6185 bp in size. M1 is the smallest minichromosome with only one gene (*rrnL*) and a 790-bp non-coding region, while M4 is the largest minichromosome with 13 genes and a 59-bp non-coding region between *nad2* and *trnM*. M3 is 5167 bp in size with eight genes and a 27-bp non-coding region between *nad3* and *trnS1*. M2 minichromosome is 3194 bp in size with only *rrnS* gene and a 2494 bp non-coding region (Figure 3, Appendix A).

Sweet et al. [21] reported that the mt genome of an undescribed species of *Myrsidea* (*Myr.* sp. 2 ex [citrine warbler]) comprised three minichromosomes. All of the mt protein-coding genes and rRNA genes we identified in *Myr.* sp. 1 ex [satin bowerbird] were also identified in the *Myr.* sp. 2 ex [citrine warbler] [21]. We identified eight mt tRNA genes in *Myr.* sp. 1 ex [satin bowerbird] and Sweet et al. [21] identified seven mt tRNA genes in the *Myr.* sp. 2 ex [citrine warbler]. A major difference between *Myr.* sp. 1 ex [satin bowerbird] and the *Myr.* sp. 2 ex [citrine warbler] is on the placement of two rRNA genes. *rrnL* and *rrnS* are on two different minichromosomes in *Myr.* sp. 1 ex [satin bowerbird] (Figure 3), while they are on the same minichromosome in *Myr.* sp. 2 ex [citrine warbler] [21]. These two species of *Myrsidea* have one minichromosome (*Y-nad5-L1-nad3-S_1_-cox2-cox1-cox3*) in common. The placement of the other eight mt protein-coding genes is the same between these two species of *Myrsidea*; however, the placement of tRNA genes associated with these protein-coding genes is different between the two *Myrsidea* species.

### 3.5. Three Species of *Laemobothrion* Retained the Typical Single-Chromosome Mitochondrial Genome Organization

We assembled the mt genomes of three species of *Laemobothrion*: *L.* sp. 1 ex [Eurasian coot], *L.* sp. 2 ex [black kite], and a currently undescribed *L.* sp. 3 ex [Australasian swamphen]. All of these species of *Laemobothrion* have the typical single-chromosome mt genomes: 14,155 bp (*L.* sp. 1 ex [eurasian coot]), 14,626 bp (*L.* sp. 2 ex [black kite]), and 14,308 bp (*L.* sp. 3 ex [Australasian swamphen]) in size, respectively (Appendix A). All of the 37 mt genes typical of bilateral animals were identified in these three species. *L.* sp. 1 ex [Eurasian coot] and *L.* sp. 3 ex [Australasian swamphen] have the same gene arrangement; all mt genes of these two species have the same orientation of transcription except *trnE*, which has an opposite orientation to other genes. The mt gene arrangement of *L.* sp. 2 ex [black kite] differs from that of the other two *Laemobothrion* species in the location of seven genes: *nad3*, *trnG*, *trnW*, *trnV*, *trnF*, *trnM*, and *trnQ* (Appendix A). Both *L.* sp. 1 ex [Eurasian coot] and *L.* sp. 3 ex [Australasian swamphen] have a 111-bp non-coding region between *trnD* and *rrnL*; *L.* sp. 2 ex [black kite] has a 327-bp non-coding region between *cob* and *nad1* (Appendix A). In contrast to the three species of *Laemobothrion* we sequenced in the current study, *L. (L.) tinnunculi* reported in Sweet et al. [21] had three minichromosomes: the *cob* gene alone was on one minichromosome; *nad2*, *trnP*, *trnG*, and *trnW* genes were on another minichromosome; and other mt genes were on the third minichromosome.

### 3.6. Phylogeny of Menoponidae Lice Inferred with Mitochondrial Genome Sequences

We reconstructed the phylogeny of Menoponidae lice (19 species from 12 genera included) with the deduced amino acid sequence of all 13 mt protein-coding genes to aid the understanding of mt genome fragmentation in this family; four species of Laemobothriidae were used as the outgroup. In both the BI and ML trees, the 19 species of Menoponidae are divided into five well-supported clades: (1) *Plegadiphilus* sp. ex [Australian white ibis], *Menacanthus cornutus* (Schömmer, 1913), and *Amyrsidea (Argimenopon) minuta* (Emerson, 1961) (all have typical single-chromosome mt genomes); (2) two *Myrsidea* species (both have fragmented mt genomes); (3) two species of *Actornithophilus* (both have fragmented mt genomes); (4) two species of *Austromenopon* (*Aus.* sp. 2 ex [sooty tern and crested tern] has a two-chromosome mt genome, whereas *Aus.* sp. 1 ex [sooty shearwater] has a single-chromosome mt genome); and (5) 10 species of Menoponidae from six genera (*Ciconiphilus*; *Colpocephalum*, *Eomenopon*, *Franciscoloa*, *Osborniella*, and *Piagetiella*; all have single-chromosome mt genomes) (Figure 4 and Figure 5). All of these five clades (Clades 1–5) are well supported in both the ML and BI trees, although the relationships among these five clades differ between the two trees (Figure 4 and Figure 5). Within the outgroup, *L.* sp. 1 ex [Eurasian coot] and the undescribed *L.* sp. 3 ex [Australasian swamphen] are closely related, both having single-chromosome mt genomes (Figure 4 and Figure 5). *Laemobothrion* sp. 2 ex [black kite], which has a single-chromosome mt genome (Appendix A), is sister to *L. tinnunculi* (Figure 4 and Figure 5), which has three mt minichromosomes according to Sweet et al. [21].

## 4. Discussion

### 4.1. Mitochondrial Genome Fragmentation Occurred Multiple Times Independently in the Species of Menoponidae and Laemobothriidae

Our results showed that while most species in the families Menoponidae and Laemobothriidae investigated to date retained the typical single-chromosome mt genomes ancestral to animals, mt genome fragmentation occurred once in the Laemobothriidae and multiple times independently in the Menoponidae: two times according to the ML tree (Figure 4 and Figure 6) and three times according to the BI tree (Figure 5). Both *Myrsidea* species investigated to date have fragmented mt genomes, and so do the two *Actornithophilus* species. For the genus *Austromenopon*, one species, *Aus.* sp. 2 ex [sooty tern and crested tern], has a fragmented mt genome, whereas the other species, *Aus.* sp. 1 ex [sooty shearwater], has a single-chromosome mt genome. This is also true for the family Laemobothriidae, which has one genus *Laemobothrion* with 20 species [1]: only *L. (L.) tinnunculi* has a fragmented mt genome, whereas three other *Laemobothrion* species have single-chromosome mt genomes. Sweet et al. [21] reported that the mt genome of *L. (L.) tinnunculi* had three mt minichromosomes: *cob* minichromosome, *nad2-P-W-G* minichromosome, and a minichromosome with all other 32 genes. This is in stark contrast to the three *Laemobothrion* species we sequenced in the current study: all 37 mt genes are on the same single chromosome in each species (Appendix A). For the genera *Myrsidea* and *Actornithophilus*, it is not clear yet whether all species in these genera have fragmented mt genomes or not. *Myrsidea* is the most species-rich genus in the Menoponidae, with 208 species; *Actornithophilus* is the fifth species-rich genus in the family with 37 species [1]. Only two species from each of these two genera have been studied to date; more species should be investigated in future studies to find out the taxonomic scale of mt genome fragmentation in these genera.

In addition to species of Menoponidae and Laemobothriidae, fragmented mt genomes were previously found in: (1) two other amblyceran species from different families, *Cummingsia maculate* (Trimenoponidae) and *Macrogyropus costalimai* (Gyropidae) [21]; (2) 17 ischnoceran species from 15 genera [19,20]; and (3) 22 species of eutherian mammal lice from three parvorders (Anoplura, Rhynchophthirina, and Trichodectera) [16]. The evidence available to date indicates that mt genome fragmentation occurred independently at least 14 times among parasitic lice (infraorder Phthiraptera): once in eutherian mammal lice in three parvorders (Anoplura, Rhynchophthirina, and Trichodectera) [16,20], four to five times in amblyceran lice ([20], and the present study), and nine times in ischnoceran lice [20]. The independent evolution of fragmented mt genomes is also supported by the observations of the typical single-chromosome mt genomes in 16 amblyceran species from four different families (Boopiidae, Laemobothriidae, Menoponidae, and Ricinidae) ([16,17,22,23], and the current study), in 10 ischnoceran species from 10 genera [16,18,21,41], but not in any species in Anoplura, Rhynchophthirina, or Trichodectera [4,5,6,7,8,9,10,11,12,13,14,15,16,17].

### 4.2. Minichromosomal Characters Derived from Mitochondrial Genome Fragmentation Provide Valuable Information for Resolving Phylogenetic Relationships at Different Taxonomic Levels

Song et al. [16] showed that mt genome fragmentation and minichromosomal characters united parasitic lice of eutherian mammals in three parvorders: Anoplura, Rhynchophthirina, and Trichodectera. All species in these three parvorders studied to date have fragmented mt genomes with a varying number of minichromosomes, from 9 to 20 in each species [4,5,6,7,8,9,10,11,12,13,14,15,17]. When only protein-coding and rRNA genes (these genes are more stable than tRNA genes in chromosomal locations) are considered, five minichromosomes, which contain *cox1*, *nad4*, *nad5*, *rrnS*, and *rrnL*, respectively, are in common among all eutherian mammal lice in these three parvorders, whereas other minichromosomes are variable [16].

In bird lice, extensively fragmented mt genomes were previously characterized in three *Columbicola* species (family Philopteridae, parvorder Ischnocera) [19]. Unlike eutherian mammal lice in Anoplura, Rhynchophthirina, and Trichodectera, these *Columbicola* species have a very similar number of mt minichromosomes: (1) 15 for *Columbicola macrourae* (Wilson, 1941) (two protein-coding genes, *atp8* and *nad6* not identified); (2) 16 for *Columbicola columbae* (Linnaeus, 1758); and (3) 17 for *Columbicola passerinae* (Wilson, 1941). Furthermore, these *Columbicola* species share the same distribution pattern of all protein-coding and rRNA genes among the minichromosomes: each protein-coding or rRNA gene has its own minichromosome that is not shared with any other protein-coding or rRNA genes [19]. The differences among these *Columbicola* species are entirely in the distribution of tRNA genes among minichromosomes, which varies between species and even between two strains of *Col. passerinae* from different host species [19]. The genus *Columbicola* has 70 species [1]. It remains to be studied whether other *Columbicola* species have fragmented mt genomes. The minichromosomal characters derived from mt genome fragmentation are potentially a valuable source of information for resolving phylogenies among *Columbicola* species.

Sweet et al. [20] recently showed that *Columbicola passerinae* (dove louse) was closely related to *Craspedonirmus immer* (common loon louse) (Emerson, 1955), which had a fragmented mt genome with seven minichromosomes. Whether these two genera, *Columbicola* and *Craspedonirmus*, share a common event of mt genome fragmentation remains to be investigated. Our comparison of the mt karyotypes between these two genera did not reveal any shared mt minichromosomal characters between them. This is also the case for another two pairs of ischnoceran genera (between *Pessoaiella* and *Osculotes*, between *Cheloptistes* and *Oxylipeurus*) and a pair of amblyceran genera (between *Cummingsia* and *Macrogyropus*) reported by Sweet et al. [20]; we did not find any shared mt minichromosomal characters between the two genera in each pair. Most of the mt minichromosomes of these genera are partially sequenced so far and most tRNA genes were not identified in Sweet et al. [20]. In addition, only one species from each of these genera has been sequenced, except for *Columbicola*. More species from each genus and more complete mt genome data from these genera are needed to better understand the evolution of fragmented mt genomes in these genera. Nevertheless, our analysis of the results reported by Sweet et al. [20] revealed that: (1) a derived *cox1-cob* minichromosome shared between two ischnoceran genera, *Penenirmus* and *Saemundssonia*; and (2) derived *cox2-nad1* and *nad5* minichromosomes shared among three ischnoceran genera, *Penenirmus*, *Quadraceps*, and *Saemundssonia*. The three later genera and *Alcedoecus* may share a common mt genome fragmentation event according to Sweet et al. [20].

The fragmented mt genomes observed in the species of Menoponidae and Laemobothriidae (parvorder Amblycera) in the current study and Sweet et al. [21] differ from those previously seen in other parvorders. First, the mt genomes of the species of Menoponidae and Laemobothriidae are less fragmented. In contrast to 9 to 20 minichromosomes in eutherian mammal lice (Anoplura, Rhynchophthirina, and Trichodectera) [16]) and 15 to 17 in *Columbicola* species (Ischnocera) [19], *Aus.* sp. 2 ex [sooty tern and crested tern] has only two minichromosomes (Figure 1), *L. (L.) tinnunculi* has three minichromosomes [21], species of *Myrsidea* have three to four minichromosomes (14 to 15 tRNA genes not identified, Figure 3, [21]), and species of *Actornithophilus* have five to six minichromosomes (Figure 2). Second, the species of Menoponidae and Laemobothriidae tend to have two or more short non-coding regions in each minichromosome; e.g., the M2 minichromosome of *Act.* sp. 1 ex [pied oystercatcher] has four non-coding regions, 86 to 832 bp in size (Figure 2). This is in contrast to the eutherian mammal lice and the *Columbicola* species, which have a single non-coding region, 1000 to 1800 bp in size, in most minichromosomes [4,5,6,7,8,9,10,11,12,13,15,19]. Third, the non-coding regions of the species of Menoponidae with fragmented mt genomes are not conserved in sequence among the different minichromosomes of each species. This is in contrast to the non-coding regions of eutherian mammal lice and *Columbicola* species, which are highly conserved in sequence among all the minichromosomes of each species [4,5,6,7,8,9,10,11,12,15,19].

Shared derived minichromosomal characters are also observed in some species of Menoponidae. Currently, only two genera of Menoponidae, *Actornithophilus* and *Myrsidea*, each have two species known with fragmented mt genomes, whereas *Austromenopon* and *Laemobothrion* each have only a single species known to date with fragmented mt genomes (Figure 4, Figure 5 and Figure 6; [21]). The two *Actornithophilus* species exclusively share one minichromosome (*E-nad4L-nad4*) and five gene clusters on other minichromosomes (*V-K-cob-nad1*, *nad6-H-S2*, *nad2-S1-F*, *Y-atp8-atp6-N*, and *G-I-cox2-cox1-C-cox3*) (Figure 2, Appendix A). The two species of *Myrsidea* also exclusively share one minichromosome (*Y-nad5-L1-nad3-S1-cox2-cox1-cox3*) and a gene cluster (*G-cob-Q-atp6-atp8-nad4L-nad4-nad6-nad1*) on a minichromosome (Appendix A). These unique shared derived characters are examples of possible synapomorphies produced by independent mt genome fragmentation events that can be further studied to help resolve the phylogenetic relationships among species within each of these genera.

## 5. Conclusions

In the current study, we sequenced the mt genomes of 17 species of bird lice in two families: Menoponidae and Laemobothriidae. We found that four species of Menoponidae (*Act.* sp. 1 ex [pied oystercatcher], *Act.* sp. 2 ex [masked lapwing], *Aus.* sp. 2 ex [sooty tern and crested tern], and *Myr.* sp. 1 ex [satin bowerbird]) have fragmented mt genomes, whereas the other 13 species retain the typical single-chromosome mt genomes. We showed that mt genome fragmentation occurred once in the Laemobothriidae and two to three times independently in the Menoponidae. More broadly, mt genome fragmentation occurred at least 14 times independently in parasitic lice: (1) once in eutherian mammal lice in the parvorders Anoplura, Rhynchophthirina, and Trichodectera [16]; (2) four to five times in the parvorder Amblycera ([20], and the current study); and (3) nine times in the parvorder Ischnocera [20]. Mitochondrial genome fragmentation is a large-scale mutation; each independent genome fragmentation event is expected to produce a unique pattern of gene distribution among minichromosomes not seen in other fragmentation events. We found derived minichromosomal characters shared between two species of *Myrsidea*, between two species of *Actornithophilus*, between two ischnoceran genera, *Penenirmus* and *Saemundssonia*, and among three ischnoceran genera, *Penenirmus*, *Quadraceps*, and *Saemundssonia*, respectively. Although mt genome fragmentation, as a general feature, does not unite all parasitic lice that have this feature, each independent mt genome fragmentation event does produce shared derived minichromosomal characters that can be informative in resolving phylogeny of parasitic lice at different taxonomic levels. Finally, why did mt genome fragmentation repeatedly occur in bird lice but only once in eutherian mammal lice in the Anoplura, Rhynchophthirina, and Trichodectera? Why did mt genome become fragmented in some bird lice but not in other bird lice? These questions are of broad significance in genome evolution and should be investigated in future studies.

## Figures and Tables

**Figure 1 animals-13-02046-f001:**
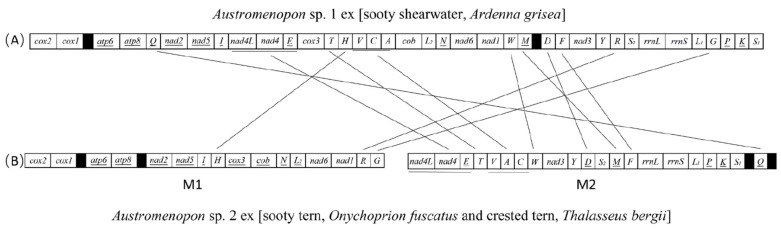
The mitochondrial genomes of (**A**) *Austromenopon* sp. 1 ex [sooty shearwater] and (**B**) *Aus.* sp. 2 ex [sooty tern and crested tern]. Gene names are: *atp6* and *atp8* for ATP synthase subunits 6 and 8; *cob* for cytochrome b; *cox1–3* for cytochrome c oxidase subunits 1–3, *nad1–5* and *nad4L* for NADH dehydrogenase subunits 1–5 and 4L; *rrnS* and *rrnL* for small and large subunits of ribosomal RNA. tRNA genes are indicated with their single-letter abbreviations of the corresponding amino acids. Genes are transcribed from left to right except those underlined, which have an opposite orientation of transcription. Non-coding regions are shaded in black. Translocated genes between *Aus.* sp. 1 ex [sooty shearwater] and *Aus.* sp. 2 ex [sooty tern and crested tern] are linked by lines. Circular minichromosomes are linearized for the sake of illustration.

**Figure 2 animals-13-02046-f002:**
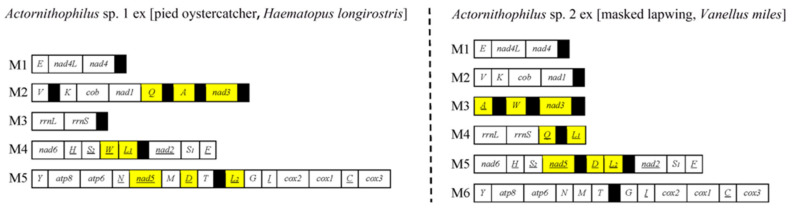
The mitochondrial genomes of *Actornithophilus* sp. 1 ex [pied oystercatcher] and *Act.* sp. 2 ex [masked lapwing]. See Figure 1 legend for description of gene names, transcription orientation, and non-coding regions. Translocated genes between these two species are shaded in yellow. Circular minichromosomes are linearized for the sake of illustration.

**Figure 3 animals-13-02046-f003:**
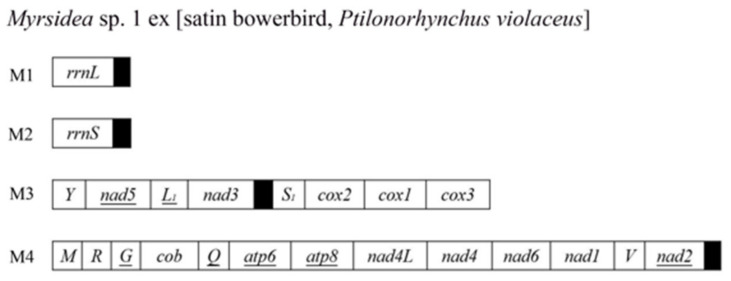
The mitochondrial genome of *Myrsidea* sp. 1 ex [satin bowerbird]. See Figure 1 legend for description of gene names, transcription orientation, and non-coding regions. Circular minichromosomes are linearized for the sake of illustration.

**Figure 4 animals-13-02046-f004:**
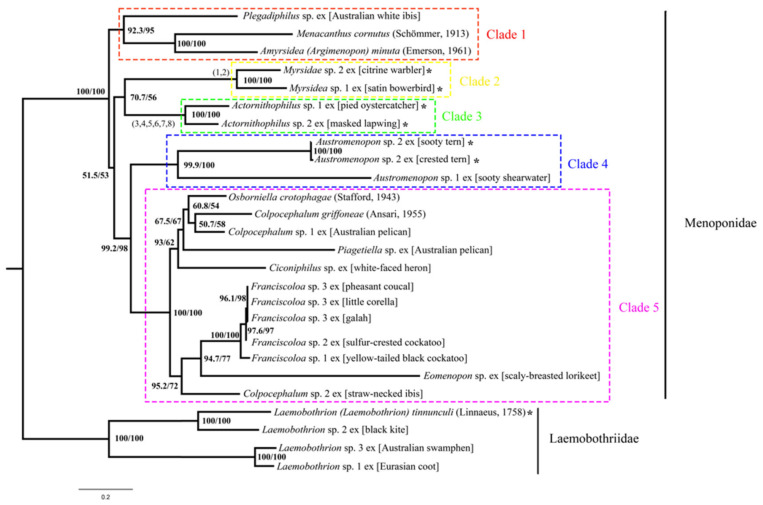
Phylogenetic relationships among 23 species of chewing lice (parvorder Amblycera) inferred from maximum likelihood analysis of deduced amino acid sequences of all mitochondrial protein-coding genes. Species of Laemobothriidae were used as the outgroup. Five clades are colored and numbered from top to bottom; these clades are well-supported in both the ML tree here and the Bayesian tree (Figure 5). Ultrafast bootstrap support values were indicated near each node of ML tree. Species with fragmented mitochondrial genomes are marked with * symbol. Numbers in brackets indicate the following shared derived mitochondrial minichromosomal characters: (1) minichromosome *Y-nad5-L1-nad3-S1-cox2-cox1-cox3*; (2) gene cluster *G-cob-Q-atp6-atp8-nad4L-nad4-nad6-nad1*; (3) minichromosome *E-nad4L-nad4*; (4) gene cluster *V-K-cob-nad1*; (5) gene cluster *nad6-H-S2*; (6) gene cluster *nad2-S1-F*; (7) gene cluster *Y-atp8-atp6-N*; and (8) gene cluster *G-I-cox2-cox1-C-cox3*. The scale bar below the tree indicates 0.2 substitutions per site.

**Figure 5 animals-13-02046-f005:**
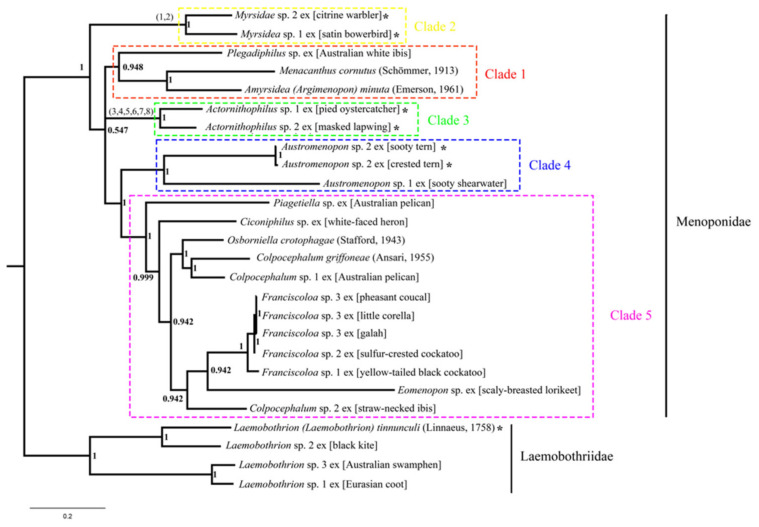
Phylogenetic relationships among 23 species of chewing lice (parvorder Amblycera) inferred from Bayesian analysis of deduced amino acid sequences of all mitochondrial protein-coding genes. Species of Laemobothriidae were used as the outgroup. Five colored clades are well-supported in both the ML tree (Figure 4) and the Bayesian tree here. Bayesian posterior probabilities were indicated near each node of the BI tree. The number of samples of the BI tree is 9002 and the effective sample size (ESS) is 8735. The mcmc convergence was checked in Tracer 1.7.2 [37]. Species with fragmented mitochondrial genomes are marked with * symbol. Numbers in brackets indicated the shared derived mitochondrial minichromosomal characters detailed in Figure 4 legend. The scale bar below the tree indicates 0.2 substitutions per site.

**Figure 6 animals-13-02046-f006:**
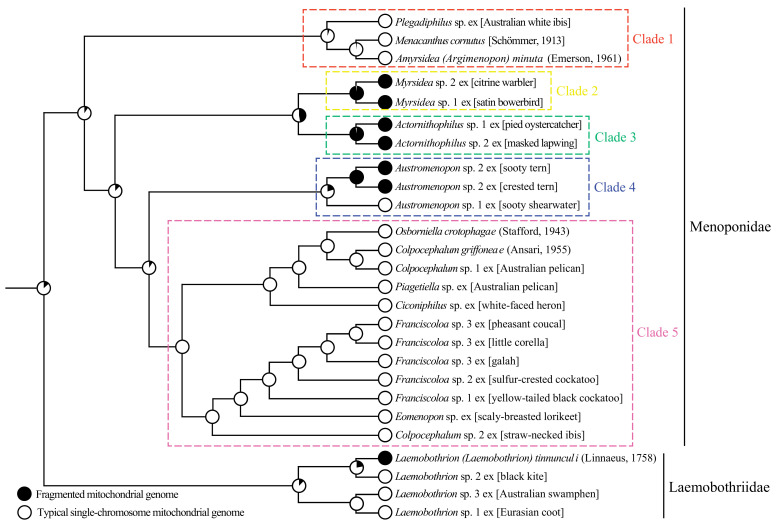
The ancestral states of mitochondrial genome organization inferred on the maximum likelihood tree (Figure 4) with Mesquite [38] using the likelihood ancestral state reconstruction method and the Mk1 model (Markov 1 parameter). Circles at the tips indicate the mitogenome structure of parasitic lice: fragmented (black) or a single chromosome (white). Pie charts at the internal nodes indicate the ancestral mitochondrial genome organisation, with the area of the black part representing the relative likelihood of the ancestral mitochondrial genome organisation being fragmented. The ancestral states inferred with the AsymmMk model (two parameters) differ only slightly in the size of the black slice relative to the white slice in several pie charts at internal nodes from those inferred with the Mk1 model.

**Table 1 animals-13-02046-t001:** Species of chewing lice included in the phylogenetic analyses in this study.

Louse Species	Host	SRA Biosample Accession Number	Number of Illumina Sequence Reads Obtained	GenBank Accession Number	References	Collection Location	Number of Specimens Used in DNA Extraction
*Actornithophilus* sp. 1	Pied oystercatcher (*Haematopus longirostris*)	SAMN29722645	19,486,216	ON417463-67	Present study	Manly, QLD *	21
*Actornithophilus* sp. 2	Masked lapwing (*Vanellus miles*)	SAMN29722639	20,263,604	ON380416-21	Present study	North Aramara, QLD	30
*Amyrsidea (Argimenopon) minuta* (Emerson, 1961)	Green peafowl (*Pavo muticus*)			MH001227	Song et al., 2019 [16]	Changsha, China	
*Austromenopon* sp. 1	Sooty shearwater (*Ardenna grisea*)	SAMN29722648	17,438,954	ON380415	Present study	Minyama, QLD	19
*Austromenopon* sp. 2	Sooty tern (*Onychoprion fuscatus*)	SAMN29722646	20,656,938	ON380413-14	Present study	Kings Beach, QLD	20
*Austromenopon* sp. 2	Crested tern (*Thalasseus bergii*)	SAMN29722643	15,659,674	ON380422-23	Present study	Caloundra, QLD	39
*Ciconiphilus* sp.	White-faced heron (*Egretta novaehollandiae*)	SAMN29722644	23,418,092	OM912467	Present study	Rothwell, QLD	31
*Colpocephalum* sp. 1	Australian pelican (*Pelecanus conspicillatus*)	SAMN29722638	12,309,898	ON204017	Present study	Amity Pt, QLD	31
*Colpocephalum griffoneae* (Ansari, 1955)	Himalayan vulture (*Gyps himalayensis*)			MH001228	Song et al., 2019 [16]	Changsha, China	
*Colpocephalum* sp. 2	Straw-necked ibis (*Threskiornis spinicollis*)	SAMN29722649	15,812,316	ON204018	Present study	Beerburrum, QLD	29
*Eomenopon* sp.	Scaly-breasted lorikeet (*Trichoglossus chlorolepidotus*)	SAMN29722636	12,065,720	ON184000	Present study	Caloundra, QLD	20
*Franciscoloa* sp. 1	Yellow-tailed black cockatoo (*Calyptorhynchus funereus*)	SAMN29722642	15,891,768	ON016537	Present study	Curramone, QLD	30
*Franciscoloa* sp. 2	Sulfur-crested cockatoo (*Cacatua galerita*)	SAMN29722641	13,324,866	ON016538	Present study	Buderim, QLD	40
*Franciscoloa* sp. 3	Little corella (*Cacatua sanguinea*)	SAMN29722635	9,794,058	ON303707	Present study	Moffat Beach, QLD	50
*Franciscoloa* sp. 3	Galah (*Eolophus roseicapilla*)	SAMN29722640	14,514,292	ON303708	Present study	Glass House mountains, QLD	30
*Franciscoloa* sp. 3	Pheasant coucal (*Centropus phasianinus*)	SAMN29722650	14,626,494	ON303709	Present study	Laceys Creek, QLD	30
*Laemobothrion* sp. 1	Eurasian coot (*Fulica atra*)	SAMN29722652	12,096,816	OM935763	Present study	Warana, QLD	1
*Laemobothrion* sp. 2	Black kite (*Milvus migrans*)	SAMN29722651	15,152,282	OM935762	Present study	Caloundra, QLD	1
*Laemobothrion* sp. 3	Australian swamphen (*Porphyrio porphyrio bellus*)	SAMN29722653	23,631,838	OM912466	Present study	Caloundra, QLD	1
*Laemobothrion (Laemobothrion) tinnunculi* (Linnaeus, 1758)	Australian hobby (*Falco longipennis*)			MW199169	Sweet et al., 2021 [21]	Australia	
*Menacanthus cornutus* (Schömmer, 1913)	Red junglefowl (*Gallus gallus*)			NC062859	Gong et al., 2022 [23]	Chongqing, China	
*Myrsidea* sp. 1	Satin bowerbird (*Ptilonorhynchus violaceus*)	SAMN29722647	25,679,092	ON417459-60, ON417462, OP271738	Present study	Maleny, QLD	20
*Myrsidea* sp. 2	Citrine warbler (*Myiothlypis luteoviridis*)			MW199172-74	Sweet et al., 2021 [21]	Peru	
*Osborniella crotophagae* (Stafford, 1943)	Smooth-billed ani (*Crotophaga ani*)			MW199175	Sweet et al., 2021 [21]	Panama	
*Piagetiella* sp.	Australian Pelican (*Pelecanus conspicillatus*)	SAMN29722637	19,903,264	ON417468	Present study	Amity Pt, QLD	14
*Plegadiphilus* sp.	Australian white ibis (*Threskiornis moluccus*)	SAMN29722634	9,680,062	ON183999	Present study	Fraser Coast, QLD	20

* QLD: Queensland, Australia.

## Data Availability

The mitochondrial genome sequence data of all the species included in this study are available in GenBank (https://www.ncbi.nlm.nih.gov/genbank/) (accessed on 9 June 2023) and the accession numbers are provided in Table 1. Cleaned Illumina sequences data the 17 Amblycera species sequenced in this study has been submitted to NCBI SRA database (Biosample accessions numbers available in Table 1).

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
