# Peer review of "Mitochondrial Genome Fragmentation Occurred Multiple Times Independently in Bird Lice of the Families Menoponidae and Laemobothriidae"

_animals, 2023, doi:10.3390/ani13122046_

Round 1

Reviewer 1 Report

This is a nice and interesting study to sequenced the mitochondrial genomes of 17 species of bird lice in the families Menoponidae and Laemobothriidae. Four species of Menoponidae have fragmented mitochondrial genomes whereas the other 13 species of both Menoponidae and Laemobothriidae retain the typical single-chromosome mitochondrial genomes. The phylogenetic analyses were also conducted and showed that mitochondrial genome fragmentation occurred multiple times independently in bird lice. The methods are comprehensive and the text is also well arranged and in good writing. I suggest to accept the manuscript at the present form.

Reviewer 2 Report

It is a nice study, clearly bringing some novel knowledge about mitogenome fragmentation, a still not-much-researched phenomenon. However, the study has one significant issue: identifying the lice species. Differences between species are one of its central points, but how the lice were identified at the species level needs to be explained. According to the methods, the lice were morphologically examined under a stereomicroscope. That usually enables determining them to genus only, so how were the species identified? I strongly suspect they were identified purely according to host associations. If this is true, it is a plague of many papers dealing with lice species and should be avoided. When you find a louse on a bird and this louse belongs to some genus, you cannot automatically assume that it is the same species as other lice of that genus which were found there in the past. First, the previous records (mentioned in Price et al.) could be misidentified. Second, you can have a new host association (there are cases when more species of one louse genus occur on one host species). Third, you obtained the lice from birds in a zoo or wildlife hospital, where there is a high probability of finding stragglers or contamination. Fourth, you used more individuals in some lice, making contamination even more probable. The correct approach is to morphologically compare the vouchers to the type material of the species you think belong to or to the respective literature. Therefore, an essential thing this manuscript needs is one of the following options: 1) If you determined the species morphologically, please explain it in more detail; 2) If you just copied the species names from Price et al. 2003 based on host association, examine the vouchers morphologically, compare them to primary sources and confirm that they actually belong to the species you claim; 3) If it is not possible to examine them, change the manuscript so that it does not include these misleading assumptions, e.g. by using genera only – for instance, “Myrsidea sp. ex [host]” or “Myrsidea sp. 1" – it would not affect the point of the manuscript. Besides this general comment, I have a few more questions about particular details; they are attached in a separate file.

Reviewer 3 Report

This is a nicely put together manuscript that further confirms that mitochondrion genome fragmentation is probably reasonably common in lice. The authors add to the case that this fragmentation has arisen multiple times. The methodology and analyses are appropriate and the writing is good. I only have a couple of statements.

Line 122 - DNA not DAN!

Presumably the bird hosts sampled are effectively a random cross-section of things that have turned up at the hospital after being injured pin the wild. Perhaps a little more about this? Also, was there any scope for cross-host contamination?

While it is important to look at the distribution of mt genome fragmentation to gain a more complete picture, it would be nice to have some thoughts at the end about where this research could/should go next, what hypotheses this might engender for these studies, and how we might use this phenomenon to test aspects of louse evolution. Also, why lice? Is it something to do with the taxon? The life history?
